# Effects of the Replacement of Pork Backfat with High Oleic Sunflower Oil on the Quality of the “Chorizo Zamorano” Dry Fermented Sausage

**DOI:** 10.3390/foods11152313

**Published:** 2022-08-03

**Authors:** Miriam Hernández-Jiménez, Iván Martínez-Martín, Ana M. Vivar-Quintana, Isabel Revilla

**Affiliations:** Food Technology, Universidad de Salamanca, E.P.S. de Zamora, Avenida Requejo 33, 49022 Zamora, Spain; miriamhj@usal.es (M.H.-J.); ivanm@usal.es (I.M.-M.); avivar@usal.es (A.M.V.-Q.)

**Keywords:** fermented meat product, backfat replacement, sunflower oil, fatty acid composition, texture, color, sensory analysis

## Abstract

The “Chorizo Zamorano” dry fermented sausage is a traditional Spanish product which is highly appreciated by consumers. This paper studies the reformulation of this product in order to improve its lipid composition and its fatty acid profile and to reduce its fat content. To achieve this, the fat used in the production of the product was partially replaced with high oleic sunflower oil in proportions of 12.5%, 20%, and 50% of the total fat content. Proximate analysis, fatty acid profiles, lipid oxidation, and sensory analysis were studied. The replacement of fat with oil showed a significant effect on the evolution of the parameters analyzed during ripening in all cases. The batches with sunflower oil presented higher levels of monounsaturated fatty acids (MUFA) and lower levels of saturated fatty acids (SFA) and a similar amount of polyunsaturated fatty acids (PUFA) to the control products. The replacement of up to 20% of oil showed no significant differences for most of the physicochemical and sensory parameters analyzed at the end of the ripening.

## 1. Introduction

Dry fermented sausages are popular and traditional meat products in several countries. These products are highly stable and can be stored without refrigeration. However, this type of product has a high fat content and a fatty acid profile of animal origin in addition to a high sodium concentration, which is why excessive intake of these products is not recommended from a health point of view [1].

The province of Zamora (Spain) is a region with a long tradition of producing a type of these sausages known as ‘Chorizo Zamorano’. This is partly due to its location and climatic conditions. The main differences from other productions of neighboring areas lie on the one hand in the careful selection of the raw material, using parts of the carcass which stand out owing to their suitability for pork production, and on the other in its high paprika content with a high proportion of spicy paprika which gives it a high stability regarding oxidation [2]. The Guarantee Mark of the “Chorizo Zamorano” aims to maintain and to safeguard all these conditions. From a nutritional point of view this product shows a high proportion of saturated fatty acids, up to 35–40% [1], and this means that the food industry is seeking alternatives so as to obtain a healthier lipid fraction. In order to do so, new formulations have been developed to reduce or replace the fat content [3]. This reduction is difficult to achieve as chopped fat has important technological functions such as freeing humidity from the internal layers of the product, a process which is necessary for unaltered fermentation [4], and the development of the aroma and taste [3]. Moreover, the nature of the fatty acids influences the quality of the product in parameters such as rancidness which alters the taste, the color, and the nutritional value [5].

To date, however, certain studies have been carried out which aim to achieve this reduction by substituting pork backfat for fats of vegetable origin. This strategy has improved the nutritional value of meat products by reducing SFA levels and increasing the content of natural antioxidants such as tocopherols [6]. The feasibility of substituting pork fat for various vegetable oils in different meat products such as raw, fermented, or cooked meat products has been investigated. The substitution of animal fat for olive oil has therefore been tested in *salchichón* [7], in Pamplona *chorizo* [8], and in Bologna-type sausages [9] and frankfurters [10] with good technological and sensory results. The use of gelled emulsion of camellia oil, which is also rich in oleic acid, in cooked sausages showed no clear changes in the properties of these sausages [11] while the use of interesterified palm kernel oil did not have a negative effect on the sensory properties of the fermented *suçuk* product [12]. Moreover, the use of canola and flaxseed oils gave high emulsion stability and sensory quality to sausages [13]. However, the use of flaxseed oil in both cooked meat products such as mortadella sausage and fermented meat products increased the PUFA content, but substantial changes in sensory properties and the drying process were observed [14,15]. Similar results were observed for *Echium* oil [16]. Regarding the use of sunflower oil to replace pork backfat, results in cooked products point to an increase in the PUFA content without any negative influences on physicochemical, textural, and sensory characteristics [17,18] while in the case of high oleic sunflower oil an increase in oleic acid was observed [19]. As for fermented dry products, satisfactory sensory characteristics were also observed in a reduced-fat *fuet* in which 5% of the pork fat was replaced by sunflower oil [20].

Reducing solid fat or replacing it with liquid oil is a huge scientific challenge which has not yet been fully resolved. The major technological problem detected when using a higher percentage of substitution is dripping fat. New proposals for liquid phase oil stabilization and structuring have recently been put forward. Different strategies such as the use of structured oil systems, interesterification, and encapsulation have been tested to solve this problem [21]. Emulsion gels (hydrogels and organogels) have been successfully used to replace/reduce animal fat in meat products [22,23].

The objective of this study is to assess the effect of the replacement of animal fat with high oleic sunflower oil in proportions of 12.5%, 20%, and 50% so as to improve its lipid profile. This oil is characterized by its low content in saturated fatty acids (8%) and by being rich in monounsaturated fatty acids (MUFA) (87%) and polyunsaturated fatty acids (5%) [24]. It has a lower percentage of saturated fatty acids and a higher percentage of monounsaturated fatty acids (MUFA) than olive oil [25]. It is also a lower-priced oil, which makes it particularly interesting for the industry. The evolution during the ripening of the physicochemical composition, the texture, and the instrumental color were studied. In addition, the composition characteristics including the fatty acid sensory characteristics were assessed at the end of the ripening period.

## 2. Materials and Methods

### 2.1. Raw Materials and the Production Process

The dry fermented sausages were prepared at a small-scale manufacturing plant in the laboratory of Food Technology in Zamora, Spain. Four samples of approximately 5 kg each of the traditional product (50% lean pork, 50% pork backfat, *w*/*w*) and four other samples of the reduced fat product (70% lean pork, 30% pork backfat) were treated. Each treatment was prepared in duplicate. For both products, a control was established together with three formulations in which pig fat was replaced with high oleic sunflower oil in proportions of 12.5%, 20%, and 50% (Table 1). The spices used per kilogram of mixture were: paprika (18.6 g of medium spicy paprika and 9.3 g of spicy paprika), 24 g of salt, 1.5 g of garlic, 1 g of oregano, and 1 g of black pepper which were purchased at local markets. The additives were as follows: 1 g of 0.1% glucose (D (+)-Glucose, Merck Eurolab, Briare Le Canal, France), 0.45 g of ascorbic acid (Panreac, Barcelona, Spain), 0.1 g of sodium nitrite (Merck, Darmstadt, Germany), 0.15 g of sodium nitrate (Merck, Darmstadt, Germany), and 0.25 g of starter. The lyophilized starter was a suspension of *Staphylococcus carnosus* and *Lactobacillus curvatus* (CHR HANSEN, Pohlheim, Germany).

The manufacturing process was carried out according to the procedure previously described [2]. At approximately 4 °C, fresh boneless pork shoulders were trimmed of visible fat and the adhering skin of pork backfat was removed. Meat and fat were chopped and minced separately (7.5 mm opening) in a meat mincer (TALSA P114K-U3, Valencia, Spain) and then mixed with the other ingredients and sunflower oil (EROSKI, Elorrio, Vizkaya, Spain) when necessary (12.5, 20, and 50% formulations) in a vacuum for 3 min (CATO AV-30, Barcelona, Spain). The sausage mixture was then stuffed (TALSA H26EA, Valencia, Spain) into natural casings (30–32 mm diameter) which had previously been treated with a solution of 1% lactic acid. The sausages were placed in a drying chamber at 12 °C at a relative humidity (RH) of 90%. After allowing the sausages to rest for a day, the temperature of the chamber was increased to 23 °C. The fermentation stage lasted 48 h. Subsequently, the sausages were kept in the ripening chamber in which the initial temperature was 15 °C and subsequently 10 °C. The initial humidity was 75%, which was then reduced to 70%. A ventilation of 0.1–0.5 m/s was carried out, which led to weight losses of 0.4–1.5% per day. The sausages were kept under these conditions for 21 days. The two trials (Traditional and Reduced fat) were performed in duplicate.

### 2.2. Instrumental Color and Texture

Two samples of each batch were taken on the first to the 21st days of the ripening process. The color parameters L* (lightness), a* (redness), and b* (yellowness) were measured by using a MiniScan XE Plus (HunterLab, Reston, Virginia, VA, USA) with a 25 mm measuring head and diffuse/8° optical geometry. CIELab parameters were calculated for the CIE illuminant D65 and 10° standard observer conditions. Three measures were taken from the surface of two slices 1.5 cm thick placed on a white surface.

A texture profile analysis (TPA) was carried out using a Universal TA-XT2i (Stable Microsystems, Surrey, England). Samples of 1 cm in height were compressed to 0.5 mm, with a time interval between the compression of 5 s. A compression plate of 50 mm in diameter was used [26]. The force–time curves were recorded at a speed of 1 mm/s (400 pps) and the maximum force and the area of the two compression peaks and the area of the negative area between the two cycles were calculated. Subsequently, the following texture parameters, hardness (g) springiness (mm), cohesiveness, gumminess (g), and chewiness (g × mm), were evaluated. Four samples of each product were analyzed.

### 2.3. Physicochemical Analysis

The pH was measured with a pH-meter (Crison Basic 20, Barcelona, Spain) using a penetration probe electrode [27]. Water activity (aw) was determined according to the ISO [28] by using Aw Sprint equipment (Novasina, Axair Ltd., Pföffikan, Switzerland) and moisture content by oven-drying in accordance with the ISO norm [28]. Nitrite content was determined spectrophotometrically (Shimadzu UV-1603, Duisburg, Germany) according to the ISO [29]. These parameters were determined in two samples of each batch which were taken on the first to the 21st days of the ripening process.

The total fat content (extractable ether) was determined by using the ISO [30] official method. Fat oxidability was measured to determine the thiobarbituric acid reactive substance (TBARS) content of the samples according to the method of Buege and Aust [31] and expressed as mg of malonaldehyde per kg of sausage. All the determinations were carried out in duplicate on the 21st day of ripening.

### 2.4. Fatty Acid Composition

The fatty acid composition of lipids was determined according to the method described by Lurueña-Martínez et al. [32] in the samples taken on the 21st day of the ripening. Intramuscular lipids were extracted by using the chloroform/methanol procedure [33]. Extracted fatty acids were methylated with 0.2 KOH in anhydrous methanol and subsequently analyzed by gas chromatography (GC 6890 N, Agilent Technologies, Santa Clara, CA, USA) using a 100 m × 0.25 mm × 0.20 µm fused silica capillary column (SP-2560, Supelco, Inc, Bellefonte, PA, USA). In total, 1 µL was injected into the chromatograph, which was equipped with a split/splitless injector and a flame ionization detector (FID). The oven temperature program was started at 150 °C followed by increases of 1.50 °C/min up to 225 °C, at which point it was maintained for 15 min. The temperature of the injector and detector was 250 °C. The carrier gas was helium at 1 mL/min and the split ratio was 20:1. The different fatty acids were identified by their retention times using a mixture of fatty acid standards (47885-U Supelco, Sigma-Aldrich, Darmstadt, Germany). Fatty acid contents were calculated by using chromatogram peak areas and were expressed as g per 100 g of total fatty acid methyl esters. All analyses were performed in duplicate.

### 2.5. Sensory Analysis

A sensory analysis was performed on the samples after 21 days of ripening by a 10-member panel trained using the Quantitative Descriptive Analysis (QDA) methodology. Of the members, 57% were men and 43% were women of between 22 and 50 years old. They were trained according to ISO [34] in the sensory profiling of dry sausages during 10 sessions. The terminology definitions and assessment techniques were agreed upon by the panelists during training. The intensity of each parameter was assessed on a 10-point scale in which 0 referred to the absence of the descriptor and 9 to the maximum intensity of the sensory characteristic. The sensory analyses were performed in the morning outside habitual eating hours in a controlled sensory analysis laboratory with individual booths under white led lights. Four samples corresponding to the four treatments of each batch were analyzed in each session.

Firstly, the panelists assessed the external appearance of the intact whole product according to the following descriptors: ease to remove the casing (the degree of easiness of removing the casing of the slice) and external defects (the degree of the appearance of defects such as wrinkles or black spots). Subsequently they assessed the internal appearance on a 10 cm section of the sausages of each treatment, cut across their axis where they measured the following parameters: color intensity (from discolored red to dark red), meat mass binding (the level at which the granules of fat and meat are united), and clean cutting (the level at which the edges of the granules of fat and meat are well defined).

The odor, flavor, and texture parameters were assessed in 1 cm-thick slices taken at 3 cm from the end of the sausages of each treatment. The parameters evaluated were: the odor intensity (the intensity of the overall odor of the sample), the rancid odor (the intensity of the rancid odor), the off-odor (the intensity of unexpected odors), the flavor intensity (the intensity of the overall flavor of the sample), the saltiness (the basic taste sensation elicited by NaCl), the pungency (a pungent sensation in the mouth and throat), the rancidness (the intensity of the rancid flavor), the off-flavor (the intensity of unexpected flavors), the hardness (the force necessary to penetrate the meat with the incisors), the chewiness (the number of times the sample must be chewed before it can be swallowed), the juiciness (the amount of juice given off by the sample when chewed), and the fatness (the fatty or lardy sensation in the mouth when chewing the sample).

### 2.6. Statistical Analysis

The significance of the effects (treatment, ripening time, and fat level) and their interaction were obtained by using General Linear Model procedures. The means and the standard error of the means (SEM) were calculated for all variables. The data obtained at the end of the ripening process were analyzed by the analysis of variance (ANOVA). The statistical significance of the fat replacement was calculated at the *p* = 0.05 level using the F-test. The Tukey test was used to test for statistically significant differences between samples. All statistical analyses were carried out using the SPSS Package 23 (IBM, Chicago, IL, USA).

## 3. Results and Discussion

### 3.1. Physico-Chemical Parameters

The evolution of the weight losses of “chorizo” throughout the ripening period is shown in Figure 1. The percentage of the inclusion of vegetable oil, the ripening time, and the fat level (*p* < 0.001) all had a significant effect. As was observed by Muguerza et al. [8], the weight losses depend on the amount of fat in such a way that the higher the fat content the lower the weight loss over the same period of time. The final weight losses therefore found in the case of the traditional formulation were 38.4%, while in the reduced fat formulation (30% of backfat) the average loss was 52.4%, a higher value than that found by Bloukas, Paneras and Fournitzis [35] (i.e., losses of 31%) while Muguerza et al. [8] obtained losses of 38.5% for similar animal fat levels. These differences may be due to the differences in relative humidity used in this study during the ripening process.

The replacement of animal fat with high oleic sunflower oil in liquid form led to reduced weight losses during ripening, with the exception of traditional production with 50% of vegetable oil. These results were probably due to the fact that the oil forms a barrier to prevent humidity losses; they agree with those of studies using vegetable oils which have not been pre-emulsified [7,35,36]. The weight losses of traditional production with 50% of vegetable oil increased owing to the loss of fat by dripping during the first week, in such a way that the weight loss was 30% compared with 14–25% for the remainder of the samples. In the case of the formulation with reduced fat (30% of total fat), during the ripening the control showed significantly lower weight losses (*p* < 0.001); however, at the end of the process no significant differences were observed between the different batches (Table 2) in agreement with the results of Bloukas et al. [35] and Muguerza et al. [8].

The pH values showed the significant effect of the ripening and treatment time but not of the fat level. Both production types (traditional and reduced fat) show an evolution (Figure 2) characterized by an initial fall owing to the action of lactic bacteria and a subsequent increase of approximately 0.2–0.3 units per day owing to the phenomena of proteolysis [2]. It should be emphasized that while in traditional production the fall was more gradual, in reduced fat production it was faster in the early days of fermentation in accordance with that observed by Bloukas et al. [35]. As for the effect of the replacement of animal fat with oil, the latter was not statistically significant in the case of traditional production as is pointed out by Muguerza et al. [8]. However, in reduced fat production the samples with added oil showed final pH values significantly lower than those of the control (*p* < 0.05) (Table 2), which agree with that indicated by Gimeno et al. [37] and Mora-Gallego et al. [38]. According to these authors, this could probably be explained by a higher concentration of free fatty acids owing to the hydrolysis of triacylglycerols in the batches with added oil.

The evolution of the water activity is shown in Figure 3. It can be observed that starting from similar values in all cases a progressive reduction occurred until the end of the ripening period (*p* < 0.001) [39]. This was slightly more marked initially in the case of traditional production than in that of reduced fat production, which means that the fat level was statistically significant (*p* < 0.001) for this parameter. Moreover, the results during ripening showed a significant effect of the level of replacement of vegetable fat in both productions (*p* < 0.001) as the control presented lower a_w_ values, while the 20% batches presented the highest values. The higher water activity values observed in the 20% substitution batches (in both the traditional and the low-fat formulations) therefore correlate with the lower weight losses observed for these batches and confirm that the lower weight loss is due to lower water loss. These differences were maintained at the end of the production process.

Humidity showed a very similar evolution to the a_w_ values; it was found that during the ripening there was a significant effect of the time of ripening, the level of replacement, and the fat level. As has been confirmed in similar studies [38], the replacement of fat with oil leads to a greater humidity in the final product as can be observed in fat-reduced production (Table 2). As mentioned above, a higher amount of fat and/or the inclusion of liquid oil form a barrier to prevent moisture losses. However, in the case of traditional chorizo the differences in the final product were not significant.

Finally, the evolution of the nitrite levels revealed a faster reduction in the case of the reduced fat samples. It was found that the values were below the detection limit as from the 4th day as previously observed by Pérez-Álvarez et al. [40], while in the case of traditional production this occurs on the 11th day. At the end of the ripening period there were no differences between the different levels of replacement or with regard to the control for this parameter.

The total fat and the oxidability of the fat were determined at the end of the ripening period (Table 2). The results showed that the batches with 50% replacement of vegetable fat presented significantly lower total fat values in both types of production, which reveals a loss of oil during ripening. There were, however, no significant differences between the control and the remainder of the batch.

A significant effect of the fat level substitution on the fat oxidability (TBARS) was observed at the end of the ripening process. The higher the percentage of animal fat substitution the higher the malonaldehyde content for the reduced fat type. This is in accordance with previous results [35] which showed that malonaldehyde levels were higher in dry sausages produced with the direct addition of olive oil. Moreover, a significant effect on the fat reduction level was observed owing to the low values of the TBARS obtained for traditionally manufactured products. Revilla and Vivar Quintana [2] had already found that in “chorizo” dry fermented sausage a significant increase in TBAR values occurs during the first days of ripening as a result of lipolysis and the oxidation of the fatty acids released, but as ripening progresses the malonaldehyde levels decrease since the compounds formed change owing to the formation of other aromatic compounds [41]. In addition, the higher the fat level the faster the evolution of the TBARS. Despite the low final levels detected, a significant difference was observed between batches in such a way that at a greater level of replacement there was a lower content in malonaldehyde. This may be due to the fact that a greater speed of oxidation means that the formation of these new aromatic compounds is faster at higher replacement levels and therefore the reduction in the malonaldehyde at the end of the period considered was more marked. Similar results had been reported by Muguerza et al. [42] when pre-emulsified soy was added.

### 3.2. Color and Instrumental Texture

Ripening had a similar but more marked effect on the color parameters in the batches of reduced fat production and these are shown in Figure 4. An initial increase was therefore observed for L* and a* (Figure 4a,b) followed by a drop.

This increase during the first few days is related to the formation of the nitrosomioglobin pigment [43]. From the seventh day of ripening the value begins to fall owing to the progressive oxidation of the paprika [44], because any color modifications observed in dry cured meat products made with paprika are mainly due to paprika color changes rather than to meat color changes [45]. Subsequently, both the traditional and reduced-fat types (Figure 4c) showed a progressive reduction in the parameter b* as also observed by Muguerza et al. [8] related to the fall in oxymyoglobin, but above all with the oxidation of the paprika as has been mentioned [45].

As for the effect of the replacement of animal fat with high oleic sunflower oil during the ripening period, in general it can be observed that the samples with the largest amounts of vegetable oil were those with the highest values of L* towards the end of the ripening period. This fact is attributed to the lower melting point of the oil, which affects the luminosity of the product [38]. Soto et al. [46] also related the luminosity of the product to humidity, in such a way that with greater humidity higher values of L* were observed. The results in the finished product (Table 3) showed that there were no significant differences between the different batches in the traditional production; in the case of reduced-fat production only the batch with a replacement of 50% presented a significantly lower level of a* than the remainder.

The evolution of the hardness (Figure 5) reveals a significant effect of the fat content in that the lower the fat the greater the hardness in the case of the control batches. Indeed, the lower the vegetable fat content of the “chorizo” the faster the hardness increased in both types of productions. This correlates with the lower loss of humidity observed in these samples for both traditional and reduced fat products.

At the end of the ripening period in traditional production, significant differences were observed between the control or 12.5% of oil substitution and the batches with high oil percentage substitution for hardness, gumminess, and chewiness; these parameters are correlated (Table 3). This result correlates with the higher humidity of these batches as the water loss during the drying process affects the final texture [47]. Similar results were obtained by Mora-Gallego et al. [38] and Muguerza et al. [1] in such a way that these parameters showed lower values in productions in which fat was replaced by oil. In the batches produced in the traditional manner, a significant reduction in adhesiveness was also observed for the higher percentage of oil replacement, which may be due to the presence of a less adhesive oily surface, while in the reduced fat batches this parameter was not affected.

### 3.3. Fatty Acid Composition

Table 4 shows the results for fatty acid composition in the samples analyzed on the last day of ripening. The fatty acid profile of high-oleic sunflower oil is characterized by the high percentage of oleic acid C18; 1 (70.7–87.4%) followed by linoleic acid C18:2ω6 (5.4–10.8), stearic acid C18:0 (3.6–4.2%), and palmitic acid C16:0 (3.2–4.1%) with a very low amount of linolenic acid C18:3ω3 (0–0.1%) [24,48].

In both (traditional and reduced fat), the most abundant fatty acids were oleic acid (C18:1) followed by palmitic acid (C16:0), stearic acid (C18:0), and linoleic acid (C18:2ω6). As was to be expected, the replacement of part of the animal fat with high oleic sunflower oil caused an increase in the concentration of oleic acid, a greater proportion in the batches with a higher replacement level, in both types of production. In the same way the palmitic acid fell significantly as it is the most abundant fatty acid in pork backfat in addition to C12:0 in traditional production. Stearic (C18:0) and linoleic (C18:2ω6) acids showed no significant changes while α-linolenic acid (C18:3) tended to decrease; the differences were, however, only significant in the case of traditional production. The changes observed in the fatty acid profile are similar to those described by other authors for raw/cured sausages when comparing batches with and without the addition of olive oil or camellia oil [1,11] as these oils are characterized by their high amount of oleic acid.

The addition of vegetable oil caused a significant increase in total monounsaturated fatty acids (MUFA) (42.9–53.3% in traditional production, 43–50.6% in reduced fat production) and a fall in total saturated fatty acids (SFA) (43.6–33.5% in traditional production, 41.6–36.6% in reduced fat production). In the case of polyunsaturated fatty acids (PUFA) there are no significant changes between the different levels of vegetable oil substitution.

### 3.4. Sensory Profile

The results of the sensory analysis of the samples at the end of the ripening period, performed by a trained panel, are shown in Table 5. They highlight the fact that in the case of reduced fat productions there were no significant differences for any of the parameters analyzed either with the control or between the different replacement percentages. This result coincides with that previously observed for products with similar or lower initial levels of pork backfat [8,42]. In the case of traditional production with a high initial level of pork fat, significant differences were observed for the casing separation parameter. The higher the percentage of replacement with sunflower oil the easier it was to separate the casing, to such an extent that in the 50% batch the casing was almost totally separated. In this traditional production a significant reduction (*p* < 0.05) in the hardness in the batches produced with sunflower oil was also observed, which agrees with that observed in the instrumental texture.

## 4. Conclusions

The substitution of pork backfat with high oleic sunflower oil resulted in lower weight losses due to lower moisture loss and high water activity; this fact was related to the lower hardness of the batches with a high percentage of backfat substitution. The drop in pH and fat oxidation was greater in the samples with high oleic sunflower oil owing to the higher intensity of lipolysis and oxidation processes. Color was not affected at the end of maturation and the parameters of hardness and ease to remove the casing were the only ones affected by the inclusion of oil in traditional processing. As for fat composition, there was a progressive and significant decrease in SFA and an increase in MUFA with no variation in PUFA with increasing substitution of animal fat by high oleic sunflower oil. The results also show that it was possible to replace up to 20% of pork fat with the direct addition of high oleic sunflower oil without significantly modifying quality parameters, the effect of substitution being less marked in the fat-reduced batches than in the traditional batches. In both types of production these 20% samples had similar sensory characteristics, although the firmness was slightly lower as a consequence of the higher moisture content. The 20% high oleic sunflower oil batch showed a more satisfactory fatty acid profile than the control with a higher MUFA and lower SFA content.

## Figures and Tables

**Figure 1 foods-11-02313-f001:**
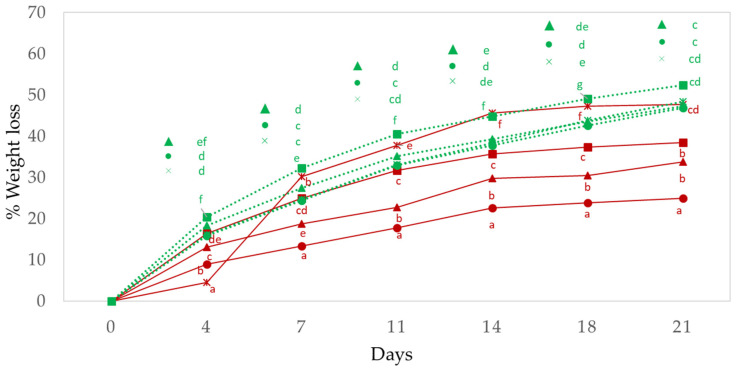
The effect of replacing pork backfat with high oleic sunflower oil on weight losses in the traditional and reduced fat “Chorizo Zamorano” during ripening. Traditional: (■) Control, (▲) 12.5%, (●) 20%, and (×) 50% of backfat replacement. Reduced fat: (■) Control, (▲) 12.5%, (●) 20%, and (×) 50% of backfat replacement. a, b, c, d, e, f, mean statistically significant differences between batches for the same day of ripening.

**Figure 2 foods-11-02313-f002:**
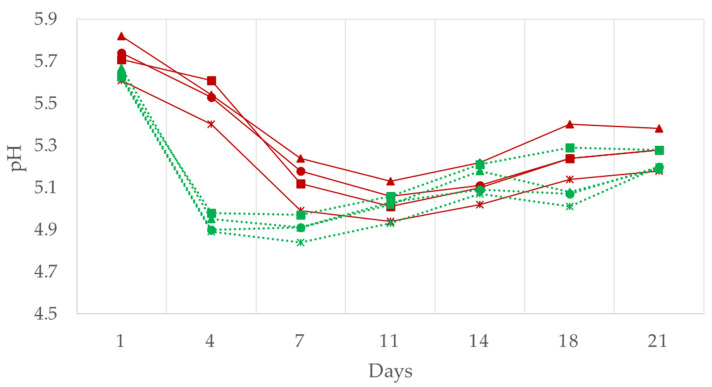
The effect of replacing pork backfat with high oleic sunflower oil on pH in the traditional and reduced fat “Chorizo Zamorano” formulations during ripening. Traditional: (■) Control, (▲) 12.5%, (●) 20%, and (×) 50% of backfat replacement. Reduced fat: (■) Control, (▲) 12.5%, (●) 20%, and (×) 50% of backfat replacement.

**Figure 3 foods-11-02313-f003:**
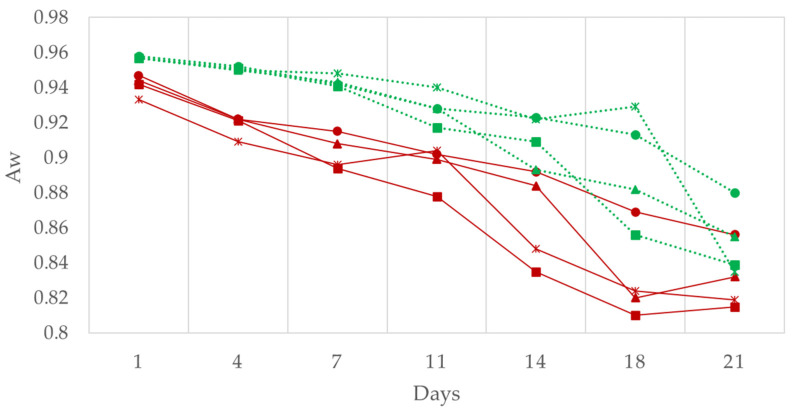
The effect of replacing pork backfat with high oleic sunflower oil on water activity in the traditional and reduced fat “Chorizo Zamorano” formulations during ripening. Traditional: (■) Control, (▲) 12.5%, (●) 20%, and (×) 50% of backfat replacement. Reduced fat: (■) Control, (▲) 12.5%, (●) 20%, and (×) 50% of backfat replacement.

**Figure 4 foods-11-02313-f004:**
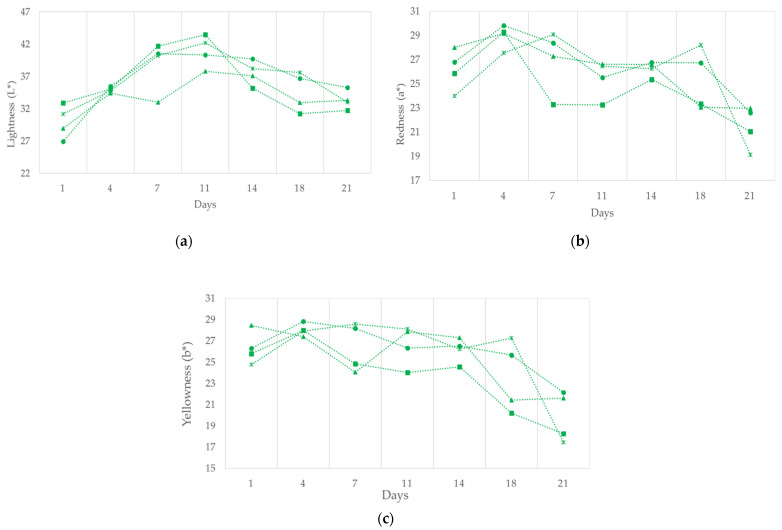
The effect of replacing pork backfat with high oleic sunflower oil on (**a**) lightness (L*), (**b**) redness (a*), and (**c**) yellowness (b*) in the reduced fat “Chorizo Zamorano” formulations during ripening. (■) Control, (▲) 12.5%, (●) 20%, and (×) 50% of backfat replacement.

**Figure 5 foods-11-02313-f005:**
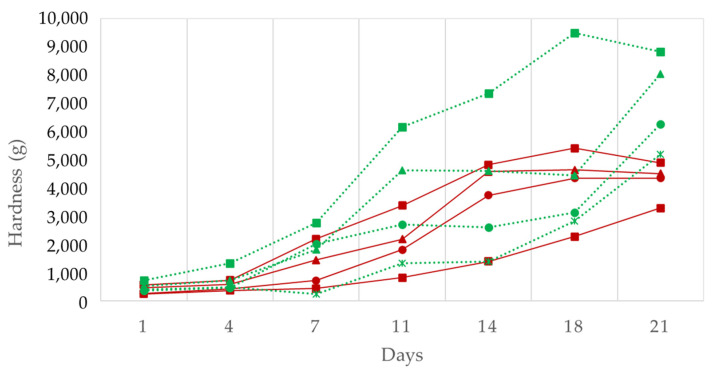
The effect of replacing pork backfat with high oleic sunflower oil on hardness in the traditional and reduced fat “Chorizo Zamorano” formulations during ripening. Traditional: (■) Control, (▲) 12.5%, (●) 20%, and (×) 50% of backfat replacement. Reduced fat: (■) Control, (▲) 12.5%, (●) 20%, and (×) 50% of backfat replacement.

**Table 1 foods-11-02313-t001:** Dry fermented sausage formulations with backfat replacement with high oleic sunflower oil as a percentage of total weight. Control: 0% replacement; 2.50% = 12.50% replacement; 20% = 20% replacement; 50% = 50% replacement.

	Traditional				Reduced Fat			
Ingredients	Control	12.5%	20%	50%	Control	12.5%	20%	50%
Lean pork	50	50	50	50	70	70	70	70
Pork backfat	50	43.75	40	25	30	26.25	24	15
High oleic sunflower oil	0	6.25	10	25	0	3.75	6	15

**Table 2 foods-11-02313-t002:** Mean values, standard deviation, and standard error of the mean (SEM) of the physico-chemical properties for the traditional and reduced fat “Chorizo Zamorano” formulations at the end of ripening.

	Traditional					Reduced Fat				
	Control	12.5%	20%	50%	SEM	Control	12.5%	20%	50%	SEM
Weight losses (%)	38.41 ± 0.49 b	33.82 ± 0.18 c	24.94 ± 0.32 d	47.75 ± 0.97 a	0.62	52.4 ± 0.62	47.2 ± 3.34	46.9 ± 0.21	48.3 ± 0.13	0.94
pH	5.28 ± 0.02	5.28 ± 0.05	5.28 ± 0.03	5.18 ± 0.02	0.03	5.28 ± 0.02 b	5.15 ± 0.04 a	5.20 ± 0.03 a	5.20 ± 0.06 a	0.01
a_w_	0.821 ± 0.009 a	0.832 ± 0.005 a,b	0.863 ± 0.001 b	0.820 ± 0.006 a	0.01	0.839 ± 0.002 a	0.855 ± 0.001 b	0.880 ± 0.002 c	0.835 ± 0.001 a	0.01
Moisture (%)	30.23 ± 0.28	29.84 ± 1.23	29.34 ± 0.12	28.98 ± 0.51	0.35	29.01 ± 1.13 a	29.8 ± 0.05 a,b	32.4 ± 0.14 b	31.0 ± 0.64 a,b	0.49
Total fat (%)	31.38 ± 0.94 b	30.45 ± 0.14 b	28.82 ± 0.34 a,b	25.45 ± 0.18 a	0.04	21.82 ± 1.01 b,c	19.7 ± 0.08 b	22.75 ± 0.22 c	16.19 ± 0.51 a	0.96
TBARS (mg MD/kg)	0.0014 ± 0.0002 b	0.0003 ± 0.0000 a,b	0.0003 ± 0.0001 a,b	0.0000 ± 0.0000 a	0.00	0.05 ± 0.001 a	0.06 ± 0.001 b	0.05 ± 0.001 c	0.06 ± 0.001 d	0.0001

Number of replicates = 6. a, b, c, d: different letters mean statistically significant differences at *p* < 0.05. MD: malonaldehyde.

**Table 3 foods-11-02313-t003:** Mean values, standard deviation, and standard error of the mean (SEM) of the color and texture properties for the traditional and reduced fat “Chorizo Zamorano” formulations at the end of ripening.

	Traditional					Reduced Fat				
	**Control**	**12.5%**	**20%**	**50%**	**SEM**	**Control**	**12.5%**	**20%**	**50%**	**SEM**
Lightness (L*)	37.99 ± 1.23	38.43 ± 0.27	38.54 ± 0.15	37.17 ± 1.21	0.46	31.73 ± 0.61	33.33 ± 0.37	35.27 ± 1.50	33.01 ± 0.44	0.53
Redness (a*)	26.29 ± 0.11	27.21 ± 0.71	27.79 ± 0.95	26.19 ± 0.34	0.54	21.04 ± 0.88 a,b	22.98 ± 0.41 b	22.6 ± 1.44 a,b	19.13 ± 0.00 a	0.62
Yellowness (b*)	26.26 ± 0.21 a	27.73 ± 0.86 a,b	28.26 ± 0.54 b	27.83 ± 0.11 a,b	0.39	18.27 ± 2.06	21.62 ± 0.11	22.16 ± 2.04	17.44 ± 0.53	0.87
	**Control**	**12.5%**	**20%**	**50%**	**SEM**	**Control**	**12.5%**	**20%**	**50%**	**SEM**
Hardness (g)	4894.86 ± 0.03 c	4513.09 ± 0.03 b	4353.36 ± 0.03 b	3289.74 ± 0.03 a	425.39	8838.19 ± 1832.03	8053.21 ± 1628.92	6275.67 ± 2390.35	5191.86 ± 1539.25	657.32
Adhesiveness (g s)	18.05 ± 10.73 a	15.08 ± 703 a	13.09 ± 5.14 a	5.40 ± 6.24 b	3.58	20.66 ± 12.93	27.81 ± 10.78	29.73 ± 24.63	20.21 ± 18.27	5.04
Springiness (mm)	0.61 ± 0.03	0.53 ± 0.00	0.56 ± 0.01	0.66 ± 0.05	0.02	0.62 ± 0.05	0.56 ± 0.04	0.52 ± 0.03	0.57 ± 0.05	0.01
Gumminess (g)	2335.81 ± 984.52 c	2286.48 ± 498.57 c	2114.61 ± 1205.30 b	1368.04 ± 745.03 a	242.48	3955.83 ± 1232.08	4286.39 ± 1145.62	3276.68 ± 1510.75	5820.70 ± 694.26	381.44
Chewiness (g mm)	1660.76 ± 269.33 c	1588.24 ± 689.15 c	1476.12 ± 575.23 b	1308.21 ± 328.32 a	174.25	2441.50 ± 568.29	2435.39 ± 777.04	1717.52 ± 872.84	1694.89 ± 218.47	234.49
Cohesiveness (g)	0.51 ± 0.03	0.56 ± 0.01	0.60 ± 0.02	0.56 ± 0.04	0.01	0.40 ± 0.02	0.48 ± 0.03	0.45 ± 0.04	0.42 ± 0.03	0.01

Number of replicates for color parameters = 6. Number of replicates for texture parameters = 8. a, b, c: different letters mean statistically significant differences at *p* < 0.05.

**Table 4 foods-11-02313-t004:** Mean values, standard deviation, and standard error of the mean (SEM) of fatty acid composition for the traditional and reduced fat “Chorizo Zamorano” formulations at the end of ripening.

	Traditional					Reduced Fat				
	Control	12.5%	20%	50%	SEM	Control	12.5%	20%	50%	SEM
C11:0	9.17 ± 0.08	0.15 ± 0.15	0.13 ± 0.04	0.11 ± 0.00	0.02	0.40 ± 0.17	0.32 ± 0.05	0.45 ± 0.12	0.17 ± 0.04	0.05
C12:0	0.10 ± 0.18 b	0.06 ± 0.30 a,b	0.03 ± 0.04 a,b	0.00 ± 0.00 a	0.01	0.21 ± 0.04	0.24 ± 006	0.06 ± 0.01	0.16 ± 0.05	0.03
C14:0	1.97 ± 0.36	1.86 ± 0.27	1.63 ± 0.19	1.34 ± 0.01	0.41	3.07 ± 1.06	2.66 ± 0.58	3.48 ± 0.77	1.72 ± 0.40	0.32
C14:1	0.15 ± 0.25	0.82 ± 0.00	0.12 ± 0.07	0.06 ± 0.00	0.02	0.25 ± 0.09 b	0.22 ± 0.02 a,b	0.06 ± 0.01 a	0.20 ± 0.00 a,b	0.03
C:15:0	0.13 ± 0.12	0.10 ±0.10	0.07 ± 0.07	0.11 ± 0.02	0.04	0.16 ± 0.05	0.22 ± 0.09	0.10 ± 0.01	0.00 ± 0.00	0.03
C16:0	28.91 ± 2.29 c	27.19 ± 2.78 c	23.42 ± 2.08 b	20.1 ± 0.18 a	0.82	26.46 ± 0.72 b	27.01 ± 0.47 b	25.52 ± 0.62 b	22.08 ± 0.78 a	0.75
C16:1	2.33 ± 1.75	1.58 ± 1.62	0.35 ± 0.04	0.31 ± 0.00	0.28	3.81 ± 0.77	3.09 ± 0.71	4.08 ± 0.74	2.32 ± 0.33	0.31
C17:0	0.35 ± 0.02	0.34 ± 0.01	0.46 ± 0.09	0.39 ± 0.00	0.05	0.41 ± 0.01	0.35 ± 0.18	0.31 ± 0.06	0.25 ± 0.11	0.04
C17:1	0.24 ± 0.01	0.26 ± 0.02	0.25 ± 0.06	0.26 ± 0.00	0.02	0.42 ± 0.18	0.24 ± 0.06	0.34 ± 0.06	0.25 ± 0.11	0.04
C18:0	11.45 ± 0.32	11.54 ± 0.19	12.54 ± 0.73	10.78 ± 0.31	0.84	10.10 ± 2.84	12.92 ± 1.44	7.47 ± 2.50	11.51 ± 0.18	0.93
C18:1t	0.39 ± 0.15 b	0.05 ±0.06 a	0.48 ± 0.04 b	0.40 ± 0.00 b	0.52	1.29 ± 1.41	0.20 ± 0.00	0.12 ± 0.00	0.11 ± 0.00	0.40
C18:1	36.87 ± 2.69 a	40.46 ± 5.54 a,b	43.46 ± 1.56 b	48.74 ± 0.37c	1.49	37.25 ± 3.27 a	38.76 ± 1.95 a,b	40.05 ± 2.68 a,b	47.77 ± 1.22 b	1.66
C18:2 (ω6)	11.56 ± 1.04	11.19 ± 0.22	10.11 ± 0.80	11.19 ± 0.29	0.82	13.37 ± 1.79	10.95 ± 0.03	15.29 ± 2.11	11.87 ± 0.40	0.72
C18:3 (ω3)	0.50 ± 0.15 b	0.57 ± 0.03 b	0.56 ± 0.06 b	0.48 ± 0.00 a	0.12	1.11 ± 0.29	0.30 ± 0.11	1.25 ± 0.33	0.74 ± 0.16	0.15
C20:3 (ω6)	0.10 ± 0.02	0.07 ± 0.00	0.13 ± 0.10	0.07 ± 0.00	0.01	0.36 ± 0.05	0.29 ± 0.09	0.21 ± 0.08	0.23 ± 0.13	0.03
C23:0	0.00 ± 0.00 a	0.00 ± 0.00 a	0.05 ±0.03 b	0.03 ± 0.00 b	0.06	0.16 ± 0.04 a,b	0.21ab ± 0.04	0.42b ± 0.07	0.08 ± 0.00 a	0.05
ΣSFA	43.55 ± 1.56 c	41.79 ± 0.54 b,c	37.42 ± 0.36 b	33.49 ± 1.14 a	1.12	41.59 ± 1.98 b	44.99c ± 0.23 c	38.60 ± 0.32 a,b	36.58 ± 0.66 a	1.23
ΣMUFA	42.88 ± 0.73 a	45.18 ± 1.24 b	50.40 ± 0.66 c	53.27 ± 2.01 c,d	1.32	43.03 ± 0.83 a	42.50 ± 1.59 a	44.65 ± 2.05 a,b	50.59 ± 1.09 b	1.28
ΣPUFA	13.32 ± 2.31	12.86 ± 1.63	11.92 ± 0.57	12.88 ± 1.89	0.93	15.34 ± 2.74	12.52 ± 1.36	16.76 ± 2.36	12.84 ± 0.44	0.84

Number of replicates = 6. a, b, c, d: different letters mean statistically significant differences at *p* < 0.05.

**Table 5 foods-11-02313-t005:** Mean values, standard deviation, and standard error of the mean (SEM) of sensory parameters for the traditional and reduced fat “Chorizo Zamorano” formulations at the end of ripening.

	Traditional					Reduced Fat				
	Control	12.5%	20%	50%	SEM	Control	12.5%	20%	50%	SEM
Easy to remove the casing	3.75 ± 2.84 a	7.60 ± 0.89 b	8.30 ± 0.67 b	9.00 ± 0.00 b	0.25	6.00 ± 2.71	6.33 ± 1.51	7.71 ± 1.11	7.86 ± 1.46	0.37
External defects	1.60 ± 2.27	3.40 ± 2.79	2.40 ± 3.33	0.00 ± 0.00	0.48	4.33 ± 3.30	3.60 ± 1.82	3.50 ± 2.65	3.50 ± 2.54	0.53
Color intensity	7.00 ± 0.94	6.00 ± 1.22	5.70 ± 1.25	6.60 ± 1.95	0.23	7.00 ± 2.89	8.29 ± 0.49	7.00 ± 2.58	7.71 ± 0.79	0.37
Meat mass binding	6.20 ± 1.68	5.20 ± 1.64	4.90 ± 2.64	4.20 ± 2.16	0.41	5.43 ± 1.39	5.14 ± 2.12	4.57 ± 2.44	4.71 ± 1.78	0.36
Clean cutting	5.20 ± 2.48	4.60 ± 1.81	4.10 ± 1.96	3.40 ± 1.81	0.25	6.43 ± 1.14	6.43 ± 1.39	5.86 ± 1.57	5.86 ± 0.98	0.23
Odor intensity	5.80 ± 1.47	5.80 ± 0.83	5.90 ± 1.59	5.40 ± 1.81	0.39	7.14 ± 1.21	6.57 ± 1.72	5.64 ± 1.75	5.64 ± 1.59	0.31
Rancid odor	0.70 ± 1.56	1.40 ± 1.67	0.40 ± 0.96	0.20 ± 0.44	0.52	2.43 ± 3.86	0.71 ± 0.95	2.14 ± 3.33	0.57 ± 1.13	0.50
Off-odor	0.10 ± 0.31	0.80 ± 0.83	0.30 ± 0.67	0.20 ± 0.44	0.32	1.43 ± 2.94	1.50 ± 2.59	1.29 ± 2.98	1.21 ± 2.07	0.48
Flavor intensity	6.80 ± 0.78	6.20 ± 1.09	6.00 ± 1.33	7.20 ± 0.44	0.56	6.71 ± 2.16	6.57 ± 2.22	6.71 ± 1.88	5.86 ± 2.38	0.39
Saltiness	4.20 ± 2.14	4.80 ± 2.68	4.50 ± 2.54	5.60 ± 1.34	0.23	4.86 ± 2.73	2.00 ± 2.08	3.71 ± 2.69	1.79 ± 1.07	0.47
Pungent	5.30 ± 1.49	4.40 ± 2.88	4.80 ± 1.98	5.80 ± 1.78	0.33	8.29 ± 0.75	7.14 ± 1.57	6.57 ± 2.37	6.57 ± 1.88	0.33
Rancidity	0.70 ± 0.94	0.60 ± 0.89	0.80 ± 1.54	0.40 ± 0.54	0.28	1.86 ± 3.33	1.43 ± 2.50	1.14 ± 2.26	1.29 ± 3.40	0.52
Off-flavor	0.20 ± 0.42	2.00 ± 3.39	1.20 ± 2.48	0.80 ± 0.83	0.62	2.57 ± 3.65	1.43 ± 2.57	1.71 ± 3.40	1.29 ± 3.40	0.59
Hardness	4.00 ± 1.63 b	2.60 ± 1.14 a,b	1.50 ± 1.26 a	2.20 ± 0.83 a,b	0.29	6.86 ± 1.46	6.57 ± 0.98	6.29 ± 1.11	5.17 ± 1.21	0.25
Chewiness	3.80 ± 2.04	2.00 ± 1.41	2.40 ± 2.27	3.20 ± 3.27	0.21	5.71 ± 1.97	5.71 ± 1.38	6.14 ± 1.67	6.43 ± 1.57	0.30
Juiciness	4.40 ± 1.64	5.80 ± 1.64	5.00 ± 1.24	4.80 ± 1.78	0.34	3.86 ± 2.11	3.71 ± 1.70	4.71 ± 1.38	3.00 ± 2.38	0.36
Fatness	3.80 ± 1.31	4.00 ± 0.70	4.50 ± 1.58	4.00 ± 1.58	0.32	3.57 ± 0.97	3.36 ± 2.13	3.43 ± 0.97	3.14 ± 2.19	0.30

Number of replicates = 20. a, b: different letters mean differences statistically significant differences at *p* < 0.05.

## Data Availability

The date are available from the corresponding author.

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
