# Peer review of "Effects of the Replacement of Pork Backfat with High Oleic Sunflower Oil on the Quality of the “Chorizo Zamorano” Dry Fermented Sausage"

_foods, 2022, doi:10.3390/foods11152313_

Round 1

Reviewer 1 Report

Oleic Sunflower oil could be considered as an interesting nutritional alternative to replace pork fat in cured meat products. Authors of manuscript investigated the fatty acid composition, physicochemical, textural and sensory characteristics. The topic of the manuscript could be considered as novel and it can provide useful information not just for the science, but also for the industry.

Generally, the manuscript is well written with a logic structure. Materials and methods are described clearly. Manuscript contains interesting and appreciated results that are discussed with relevant references.

Comments:

-        In the introduction, I suggest the authors to give more detailed discussion of the use of edible oils as fat replacers for meat industry.

-        The quality and visibility of Figures are low. Please enhance the quality of figures.

-        Please review the letters of the statistical analysis, apparently not all of them appear in the tables.

-        Line 289, please verify the name of the author 9

The conclusion is weak, please clarify the impact of the fat replacement by the oil.

Author Response

Reviewer 1

The authors are grateful for the reviewer's comments. Changes made according to the reviewer's suggestions are highlighted in blue in the text.

Comments:

  1. In the introduction, I suggest the authors to give more detailed discussion of the use of edible oils as fat replacers for meat industry.

The introduction has been expanded with special emphasis on the use of edible oils as fat replacers.

  1. The quality and visibility of Figures are low. Please enhance the quality of figures.

The figures have been improved

  1. Please review the letters of the statistical analysis, apparently not all of them appear in the tables.

We have checked this and now all the letters that appear in the table are also in the table footnote.

  1. Line 289, please verify the name of the author 9

The reference 9 corresponds to: Muguerza, E.; Fista, G.; Ansorena, D.; Astiasaran, I.; Bloukas, J.G. Effect of Fat Level and Partial Replacement of Pork Backfat with Olive Oil on Processing and Quality Characteristics of Fermented Sausages. Meat Science 2002, 61, 397–404, doi:10.1016/S0309-1740(01)00210-8.

And the name has been added to the text.

The conclusion is weak, please clarify the impact of the fat replacement by the oil.

The conclusion has been re-written to clarify the impact of the fat replacement.

Reviewer 2 Report

The manuscript entitled 'Effects of the replacement of pork backfat with high oleic sunflower oil on the “Chorizo Zamorano” dry fermented sausage qualitys' measured the adding of sunflower oil in“Chorizo Zamorano”dry fermented sausage. Overall, the content of the manuscript is clear and well written. 

1. How many replications in your study. In all the Tables and Figures, we did not find the number of replicates in this experiment and the standard variance of data.

2. In line 186, the p<0.001 was mentioned. However, significance analysis is not marked in the figure 1.

3. In line 199, “The replacement of animal fat with high oleic sunflower oil in liquid form led to reduced weight losses during ripening ”. Is there significant differences between backfat replacement groups and control. Please illustrate details in this part.

4. In line 223, please indicate the reason for the lower pH values of reduced fat groups than control.

5. In line 232, the  In”should be deleted

6. In line 236-239, 20% batches presented the highest values in the day 21, the times for ripening should be indicated here.  ...  in agreement with that observed for weight losses  what the meaning here. The weight loss of 20% batches in reduced fat group was shown in Figure 1, but the comparison was not clear. Please have an illustration on this matter.

7. The unit of TBARS values in Table 2 was lost.

8. In the part of “Fatty acid composition”, the composition or major fatty acids of oleic sunflower oil should be indicated firstly. In line 338-339, the references here were not oleic sunflower oil but others. What is the reason for this similar results.

Author Response

Reviewer 2

The authors are grateful for the reviewer's comments. Changes made according to the reviewer's suggestions are highlighted in red in the text.

  1. How many replications in your study. In all the Tables and Figures, we did not find the number of replicates in this experiment and the standard variance of data.

Each mean is the average of two trials (this point has been clarified in the line 71) and three replicates for each trial (n=6) excepting for texture parameters for which four replicates (n=8) and for sensory analysis for which ten replicates (n=20) were carried out. The number of replicates has been included in the tables as so as the standard deviation of the data

  1. In line 186, the p<0.001 was mentioned. However, significance analysis is not marked in the figure 1.
  2. In line 199, “The replacement of animal fat with high oleic sunflower oil in liquid form led to reduced weight losses during ripening ”. Is there significant differences between backfat replacement groups and control. Please illustrate details in this part.

The existence of significant differences between the backfat groups and control has been illustrated by including letters that mean statistically significant differences at p<0.05 at each point of ripening. Traditional and reducing fat formulations have been analysed together in order to highlight the existence of significant differences also between these two types of formulations, as previously requested by the reviewer.

  1. In line 223, please indicate the reason for the lower pH values of reduced fat groups than control.

The reason why the oil-added batches had a lower pH than control for reduced fat formulation has been added.

  1. In line 232, the  “In”should be deleted

It has been deleted

  1. In line 236-239, 20% batches presented the highest values in the day 21, the times for ripening should be indicated here. “ ...  in agreement with that observed for weight losses ” what the meaning here. The weight loss of 20% batches in reduced fat group was shown in Figure 1, but the comparison was not clear. Please have an illustration on this matter.

This point has been better explained in the text.

  1. The unit of TBARS values in Table 2 was lost.

It has been added

  1. In the part of “Fatty acid composition”, the composition or major fatty acids of oleic sunflower oil should be indicated firstly. In line 338-339, the references here were not oleic sunflower oil but others. What is the reason for this similar results.

The main fatty acids of high oleic sunflower oil have been added. The reason for similar results observed when other oils were added has been added

Reviewer 3 Report

Manuscript ID: foods-1825331

In the article entitled: “Effects of the replacement of pork backfat with high oleic sun flower oil on the “Chorizo Zamorano” dry fermented sausage  quality” was studies the physico-chemical and textural properties of dry fermented sausage.  In analyzed dry fermented sausage  part of the pork fat was replaced with sun flower oil.

It is an interesting article. The work have practical application. It can contribute to obtaining a healthier product with a reduced content of saturated fats. From a methodological point of view, the article uses measurement techniques appropriate to the assumed purpose of the research.

Title

The title and the aim of the study are clearly constructed.

Abstract

The abstract includes the aim of the study, methods used in the experiment and contain the principal results and conclusions.

Introduction

The introduction describes the matter of the experiment and states the problem being investigated. However, but there are no references to research from the last few years (is one of 2019). They correctly interpreted and described the significance of the results for the research. They skillfully referred to the results of other researchers.

Methods

The data is well collected. The methods as far as possible described in detail (below I have questions). The sampling is appropriate and adequately described. Statistical analysis of measurement results has been used.

Instrumental color and texture

A double compression cycle test was carried out? If so, then:

What was the compression distance?

What was the time interval between the two compressions?

What was the data acquisition?

This information should be included in the text.

Results and discussion

Generally, the authors correctly interpreted and described the significance of the results for the research. However, I feel a certain lack of satisfaction. I miss at least a short (one, two sentences) explanation of the observed effect (paragraph 240-244; 317-324).

Maybe on the drawings place error bars?

Conclusion

The authors correctly indicate, how the results are related to the studies.

Language

The article is correctly written. English language and style are minor spell check required

Author Response

Reviewer 3

The authors are grateful for the reviewer's comments. Changes made according to the reviewer's suggestions are highlighted in green in the text.

Introduction

The introduction describes the matter of the experiment and states the problem being investigated. However, but there are no references to research from the last few years (is one of 2019).

According to Reviewer 1 suggestions the introduction has been updated and more recent references has been added.

Methods

The data is well collected. The methods as far as possible described in detail (below I have questions). The sampling is appropriate and adequately described. Statistical analysis of measurement results has been used.

Instrumental color and texture

A double compression cycle test was carried out? If so, then:

What was the compression distance? 5 mm

What was the time interval between the two compressions? 5 seconds

What was the data acquisition? The force-time curves were recorded at a speed of 1 mm/s (400 pps) and the maximum force and the area of the two compression peaks and the area of the negative area between the two cycles were calculated.

This information should be included in the text.

The information has been added to the text.

Results and discussion

Generally, the authors correctly interpreted and described the significance of the results for the research. However, I feel a certain lack of satisfaction. I miss at least a short (one, two sentences) explanation of the observed effect (paragraph 240-244; 317-324).

A short explanation has been added in both paragraphs.

Maybe on the drawings place error bars?

We have not added this because some other changes have been made in the figures according to other reviewer and the resultant figures were not very clear.

Round 2

Reviewer 2 Report

All the comments have been modified rightly.  In addition, there is one point that needs to be changed. The replications in the experiment could be marked at the bottom of the Table 1-4.

Author Response

The authors are grateful for the reviewer's comments and changes made according to the reviewer's suggestions are highlighted again in red in the text.

The number of replicates of each experiment has been added at the bottom of all the tables

The manuscript has been revised by the professional translator Stephen A. Trott.